# Free Cholesterol Affects the Function and Localization of Human Na^+^/Taurocholate Cotransporting Polypeptide (NTCP) and Organic Cation Transporter 1 (OCT1)

**DOI:** 10.3390/ijms23158457

**Published:** 2022-07-30

**Authors:** Jessica Y. Idowu, Bruno Hagenbuch

**Affiliations:** Department of Pharmacology, Toxicology, and Therapeutics, The University of Kansas Medical Center, Kansas City, KS 66160, USA; jidowu@kumc.edu

**Keywords:** NTCP, OCT1, transporter, lipid raft, cholesterol, methyl-β-cyclodextrin, cholesterol-methyl-β-cyclodextrin

## Abstract

Non-alcoholic fatty liver disease (NAFLD) and non-alcoholic steatohepatitis (NASH) are associated with obesity. They are accompanied by increased levels of free cholesterol in the liver. Most free cholesterol resides within the plasma membrane. We assessed the impact of adding or removing free cholesterol on the function and localization of two hepatocellular uptake transporters: the Na^+^/taurocholate cotransporting polypeptide (NTCP) and the organic cation transporter 1 (OCT1). We used a cholesterol–MCD complex (cholesterol) to add cholesterol and methyl-β-cyclodextrin (MCD) to remove cholesterol. Our results demonstrate that adding cholesterol decreases NTCP capacity from 132 ± 20 to 69 ± 37 µL/mg/min and OCT1 capacity from 209 ± 66 to 125 ± 26 µL/mg/min. Removing cholesterol increased NTCP and OCT1 capacity to 224 ± 65 and 279 ± 20 µL/mg/min, respectively. In addition, adding cholesterol increased the localization of NTCP within lipid rafts, while adding or removing cholesterol increased OCT1 localization in lipid rafts. These results demonstrate that increased cholesterol levels can impair NTCP and OCT1 function, suggesting that the free cholesterol content of the liver can alter bile acid and drug uptake into the liver. This could explain the increased plasma bile acid levels in NAFLD and NASH patients and potentially lead to altered drug disposition.

## 1. Introduction

A hallmark of the rising obesity epidemic is the presence of increased free cholesterol levels that are associated with non-alcoholic fatty liver disease (NAFLD), non-alcoholic steatohepatitis (NASH), and altered N-glycosylation [1,2]. Given the increased incidence of obesity and fatty liver disease affecting the liver and potentially drug disposition, it is necessary to understand the impact that increased cholesterol levels have on transporters individually and in concert with each other.

Our lab has previously demonstrated that transporters interact with each other and that these interactions can alter surface expression and transporter function [3,4]. Colocalization in lipid rafts is a potential explanation for these protein–protein interactions [5,6]. Lipid rafts are clusters of glycosphingolipids, cholesterol, and proteins within the plasma membranes. They possess a dynamic structure resulting in changing lipid and protein content, with alterations in cholesterol levels potentially impacting the activity of raft-associated proteins [7,8]. There are a variety of lipid raft markers that are used and accepted across different species. Positive lipid raft markers are caveolin, associated with caveolae, and flotillin-1, associated with clathrin-independent endocytosis [9,10,11,12,13,14]. Clathrin is a well-accepted negative lipid raft marker [10,11,13]. Na^+^/K^+^ ATPase can act as either a positive or negative marker depending on the cell type [11,15,16,17,18,19]. 

Recent publications have demonstrated that transporters associated with lipid rafts undergo compartmentalization and conformational changes [11,20,21,22]. This observation can be explained by the presence of free cholesterol in the plasma membrane that can affect membrane fluidity. It also emphasizes the importance of examining cholesterol’s impact on transporters and lipid rafts. The Na^+^/taurocholate cotransporting polypeptide (NTCP) and the organic cation transporter 1 (OCT1) are hepatic uptake transporters located on the basolateral membrane of hepatocytes. They are essential for bile acid homeostasis and the disposition of antidiabetic drugs. NTCP is a sodium-dependent bile acid transporter involved in the enterohepatic circulation. It mediates the uptake of conjugated bile acids and selected drugs from portal blood into hepatocytes [23]. OCT1 is highly expressed in human hepatocytes, where it functions as an uptake transporter and mediates the first step of liver drug elimination for numerous cationic drug substrates, including metformin [24,25]. 

A previous study demonstrated that cholesterol affected the transport and localization of rat NTCP when expressed in HEK293 cells. Uptake increased while membrane localization of the protein decreased after depletion and returned to control conditions after cholesterol was replenished. However, uptake beyond a single time point and at a single taurocholate concentration was not investigated [11]. Thus far, no such studies have reported using human NTCP or OCT1. The goals of the present study were to extend these previous findings to human NTCP, compare them to OCT1, and further characterize how transport kinetics are affected when loading or depleting cholesterol by measuring uptake kinetics instead of looking at a single taurocholate concentration. Due to the difficulty of obtaining quality human hepatocytes in the necessary quantities, we used HEK293 cells for this study. HEK293 cells are easy to maintain and have been used extensively to characterize transporter expression and function. We expect these extended results to indicate whether such changes in free cholesterol could impact drug disposition. 

## 2. Results 

### 2.1. Functional Consequences of Changing Free Cholesterol Levels in the Plasma Membrane 

To investigate whether the free cholesterol content of the plasma membrane affects the function of human NTCP and OCT1, we added or removed free cholesterol from transporter-expressing HEK293 cells. We then measured the uptake of 100 μM taurocholate for NTCP and 16.7 nM methyl-4-phenylpyridinium iodide (MPP^+^) for OCT1 after a 30 min treatment with either a cholesterol–MCD complex (cholesterol) to increase or a methyl-β-cyclodextrin (MCD) to decrease free cholesterol. The results in Figure 1A demonstrate that increasing cholesterol results in a dose-dependent decrease of NTCP-mediated taurocholate uptake compared to the control (Figure 1A). Similarly, OCT1-mediated uptake decreased compared to the control in a dose-dependent manner (Figure 1B). The effects of removing free cholesterol from the plasma membrane are demonstrated in Figure 2A. For human NTCP, we observed a concentration-dependent increase in taurocholate uptake. This is consistent with the previous studies that used rat NTCP and found that cholesterol depletion increased taurocholate uptake. In contrast, for OCT1, we demonstrated a concentration-dependent decrease in MPP^+^ uptake.

To ascertain that treatment with cholesterol or MCD resulted in an increase or decrease in membrane free cholesterol, we measured the free cholesterol content in the transporter-expressing HEK293 cells using the Amplex Red Cholesterol Assay Kit in the absence of cholesterol esterase. Under these conditions, it determines free (non-esterified) cholesterol. To determine total cholesterol, cholesterol esterase would have to be added. The results are summarized in Figure 3. Based on the significant functional effects observed on both NTCP and OCT1 as shown in Figure 1 and Figure 2, treatment of NTCP-expressing cells with 0.8 mM cholesterol or 5 mM MCD, and OCT1-expressing cells with 0.4 mM cholesterol and 20 mM MCD were selected. The results demonstrate that treatment with cholesterol increased and treatment with MCD decreased the plasma membrane free cholesterol content (Figure 3). In NTCP-expressing HEK293 cells treated with 0.8 mM cholesterol, free cholesterol increased by about 60%, while treatment with 5 mM MCD decreased free cholesterol by approximately 30% (Figure 3A). In OCT1-expressing HEK293 cells, treatment with 0.4 mM cholesterol increased free cholesterol levels by almost 70%, and 20 mM MCD decreased free cholesterol by 70% (Figure 3B). These results demonstrate that the functional changes seen in Figure 1 and Figure 2 correlate with the altered plasma membrane free cholesterol content.

### 2.2. Characterization of the Functional Consequences of Altered Free Cholesterol Levels

Changing the plasma membrane cholesterol content can affect lipid rafts and raft-mediated endocytosis, which may alter NTCP and OCT1 plasma membrane expression. We performed surface biotinylation experiments to investigate how human NTCP or OCT1 plasma membrane expression are affected by adding or removing free cholesterol. The surface biotinylation was quantified using western blots with the Na^+^/K^+^ ATPase α1 subunit acting as a loading control. A 30 min treatment with 0.8 mM cholesterol or 5 mM MCD decreased NTCP surface expression by about 25% and 40%, respectively (Figure 4A). These changes were qualitatively similar to those reported by Molina et al., 2008 [11], but less pronounced. For OCT1, differences were more pronounced. Surface expression decreased by about 40% and 70%, respectively (Figure 4B). A representative western blot for NTCP is shown in Figure 4C. Some of the changes are due to correcting for the loading control, Na^+^/K^+^ ATPase. The representative OCT1 blot (Figure 4D) also demonstrates that MCD treatment significantly increased the amount of non-glycosylated OCT1 at the plasma membrane. Next, we normalized the uptake from Figure 1 and Figure 2 for the respective transporters’ surface expression. Consequently, cholesterol treatment decreased NTCP-mediated taurocholate transport by about 50% while not affecting OCT1-mediated MPP^+^ uptake (Figure 5). In contrast, after normalizing for plasma membrane transporter expression, both NTCP- and OCT1-mediated uptake were significantly increased after MCD treatment (Figure 6).

To further characterize the functional changes and determine to what extent K_m_ and V_max_ values were affected by the addition or removal of cholesterol, the initial linear portions of taurocholate and MPP^+^ uptake were determined. Based on these experiments, we performed NTCP-mediated taurocholate and OCT1-mediated MPP^+^ uptake at 15 s, well within the initial linear portion of uptake. As shown in Figure 7 and summarized in Table 1, the kinetic parameters of human NTCP-mediated taurocholate uptake were modified after treatment with cholesterol or MCD compared to control conditions. Under control conditions, NTCP-mediated taurocholate uptake was saturable with a K_m_ of 18 ± 2.8 µM and a V_max_ of 2.3 ± 0.04 nmol/mg/min, yielding an overall capacity of 132 ± 20 µL/mg/min. Following treatment with 0.8 mM cholesterol, the K_m_ increased to 38 ± 28 µM while the V_max_ decreased to 1.9 ± 0.1 nmol/mg/min, significantly reducing transport capacity to 69 ± 37 µL/mg/min. Treatment with 5 mM MCD increased both the K_m_ and the V_max_ to 39 ± 13 µM and 8.3 ± 1.0 nmol/mg/min, respectively, while also significantly increasing capacity to 224 ± 65 µL/mg/min (Figure 7 and Table 1). Similar results were obtained for OCT1-mediated uptake of MPP^+^. OCT1-mediated MPP^+^ uptake under control conditions was saturable with a K_m_ of 39 ± 11 µM and a V_max_ of 7.6 ± 1.7 nmol/mg/min, resulting in a capacity of 209 ± 66 µL/mg/min. Treatment with 0.4 mM cholesterol increased both the K_m_ and the V_max_ values to 72 ± 17 µM and 9.0 ± 2.3 nmol/mg/min, respectively, while decreasing the capacity to 125 ± 26 µL/mg/min. Following treatment with 20 mM MCD, the K_m_ decreased to 26 ± 5 µM while the V_max_ slightly decreased to 7.2 ± 1.2, and the overall capacity significantly increased to 279 ± 20 µL/mg/min (Figure 8 and Table 1).

### 2.3. Impact of Altered Membrane Cholesterol on NTCP and OCT1 Lipid Raft Localization 

As previously mentioned, altering plasma membrane cholesterol content may affect lipid raft composition. To investigate whether NTCP and OCT1 would colocalize with lipid raft markers, we analyzed samples of the 1 mL fractions collected from a discontinuous sucrose gradient separating membrane proteins from either NTCP- or OCT1-expressing HEK293 cells. Appendix A shows the sucrose concentration, the protein distribution, and the distribution of the lipid raft marker flotillin-1. Western blots were probed for NTCP, OCT1, or lipid raft markers. The gradient from NTCP-expressing HEK293 cells contained thirteen fractions collected from the top of the gradient (fraction 1) to the bottom (fraction 13). The lipid raft fractions 5–7 were visible as a cloud at the interface of the 5 and 35% sucrose concentration and are positive for flotillin-1 in all three conditions: control, cholesterol- and MCD-treatment (Figure 9). In the control condition, most NTCP does not colocalize with the lipid raft markers (Figure 9A), although some NTCP signals can be seen in fraction 6. However, adding cholesterol increased the amount of NTCP in lipid rafts despite the majority still residing outside lipid rafts (Figure 9B). Consequently, removing cholesterol also removes any detectable NTCP from the lipid raft fractions (Figure 9C). 

Results from the OCT1-expressing HEK293 sucrose gradient experiments are summarized in Figure 10. Here we analyzed 12 fractions, with fraction 1 at the top of the gradient and fraction 12 at the bottom. The lipid raft fractions represented by the sucrose interface and flotillin-1 signals are fractions 6 and 7. Under control conditions, only very faint signals of OCT1 are seen in these fractions (Figure 10A). However, upon treatment with cholesterol, the amount of OCT1 present in fractions 5–7 increased significantly (Figure 10B), and a similar trend was seen following MCD treatment (Figure 10C).

## 3. Discussion

This study aimed to extend previous findings on how changes in free cholesterol affected rodent NTCP [11] by measuring this effect in human NTCP, and comparing it to human OCT1. Here, we demonstrated that increasing cholesterol decreases the capacity of NTCP to transport taurocholate and increases its localization within lipid rafts. Furthermore, we have shown that reducing cholesterol increases NTCP function and capacity while removing it from lipid rafts. These results align with the findings of Molina et al., 2008 regarding rodent NTCP [11], and suggest that the function of NTCP inversely correlates with the free cholesterol concentration in the plasma membrane. However, additional experiments in the future are needed to evaluate the exact underlying mechanism of this phenomenon. For example, does the function change as a result of NTCP’s localization in lipid rafts, or due to cholesterol-induced changes in membrane fluidity? Alternatively, does cholesterol directly bind to the transporter and induce a conformational change in NTCP? The recently resolved structure of human NTCP has two cholesterol molecules bound to the outside of the transporter, supporting the latter hypothesis which requires additional future experiments [26]. On a more physiological or pathophysiological level, our findings suggest that increased free cholesterol levels, as seen in subjects with NAFLD and NASH [2], could result in diminished bile acid uptake into hepatocytes. Patients with NAFLD and NASH have increased plasma bile acids, mainly conjugated bile acids such as glycocholate and taurocholate, two known substrates of NTCP [27,28,29,30,31]. Thus, the reduced NTCP capacity due to increased free cholesterol levels could be partially responsible for this effect.

Concerning OCT1, we demonstrated that increasing free cholesterol decreased the capacity of OCT1 to transport its model substrate MPP^+^, and similar to NTCP, OCT1 localization in lipid rafts was increased. However, in contrast to NTCP, when decreasing cholesterol levels, OCT1 function is decreased while capacity and lipid raft colocalization is increased. This supports the idea that OCT1 has optimal functionality when the plasma membrane has baseline cholesterol content, and any alteration increases lipid raft localization and decreases function. Again, future experiments are necessary to elucidate whether the underlying mechanism is related to membrane fluidity, localization in lipid rafts, or a combination of both.

Regarding pharmacology and OCT1-mediated targeting of metformin to hepatocytes, we assume that metformin is handled similarly to MPP^+^ by OCT1 [32]. Therefore, our results postulate that patients with increased free cholesterol in the liver need dose adjustments when treated for type II diabetes with metformin because OCT1 function is diminished. 

In addition to looking at the functional impact of plasma membrane cholesterol, we evaluated the associated expression changes in both NTCP and OCT1. The surface expression of both transporters decreased after removing free cholesterol from the plasma membrane. These findings signify that having baseline cholesterol levels allows for more of the transport protein to succeed in reaching the cell’s surface or to reside at the membrane for a longer time. This is consistent with previous studies that have demonstrated that depleting the plasma membrane of free cholesterol by treatment with MCD can block clathrin-dependent endocytosis [33], and can also increase internalization of another transmembrane protein, the acetylcholine receptor [34]. However, this is another topic in need of further investigation.

The increase in NTCP function after removing free cholesterol seems to reach a plateau, suggesting that removing cholesterol beyond a certain level has no impact (Figure 2A). The functional changes observed in our study with human NTCP accord with results obtained using rat NTCP. However, the functional increase after 10 mM MCD treatment is lower in our experiments than in those that used rat NTCP [11]. 

The majority of NTCP and OCT1 were located outside of lipid rafts under our experimental conditions, and only a small amount of these proteins localized with lipid rafts under control conditions (Figure 9 and Figure 10). Increasing cholesterol increased NTCP in lipid rafts, while decreasing cholesterol completely removed it from lipid rafts (Figure 9). These findings align with those of studies that used rat NTCP, suggesting a similarity between the mechanism in humans and in rodents, and thus the potential for using an animal model to investigate the underlying mechanisms further [11]. Localization of OCT1 with lipid rafts increases whether plasma membrane cholesterol is increased or decreased in HEK293 cells (Figure 10) and correlates with reduced function, supporting the idea that residence inside lipid rafts is detrimental to the function of OCT1. However, due to the limitations of our methods, we may have underestimated the number of transporters in the respective detergent-resistant membranes, and newer techniques like super-resolution imaging will be required to obtain more definitive results [35].

A shortcoming of our study is that we only assessed 30-min changes in free cholesterol instead of changes over 24 or 48 h. In addition, we used HEK293 cells instead of human hepatocytes. Future studies would ideally be performed in human hepatocytes isolated from patients with different levels of NAFLD, in animal models of NAFLD such as the ob/ob mice, or in cell lines that are closer to hepatocytes, e.g., HepaRG or HepG2 cells that express the respective transporters [36].

In conclusion, changing the membrane free cholesterol content alters the function and localization of both NTCP and OCT1, suggesting that individuals with liver conditions like NAFLD and NASH have impaired transporter capacities. Impaired NTCP capacity compromises bile acid uptake into hepatocytes and partially explains the increased bile acids associated with NAFLD and NASH. The reduced OCT1 capacity will likely increase drug elimination through the kidneys and reduce the ability of drugs such as metformin to reach their target within hepatocytes. Future studies in more liver-like systems and with longer cholesterol exposure should allow for better delineation of the underlying mechanisms, and help to ultimately explain alterations in bile acid and drug uptake in obese patients.

## 4. Materials and Methods

### 4.1. Experimental Materials

Radiolabeled [^3^H]-Taurocholic acid was purchased from PerkinElmer (Boston, MA, USA). Radiolabeled [^3^H]-Methyl-4-phenylpyridinium iodide (MPP^+^) was obtained from American Radiolabeled Chemicals, Inc. (St. Louis, MO, USA). Methyl-β-cyclodextrin (MCD) and water-soluble Cholesterol-methyl-β-cyclodextrin (cholesterol) were acquired from Sigma Aldrich (St. Louis, MO, USA). Taurocholic acid sodium salt (97% pure) was purchased from Sigma Aldrich.

Rabbit anti-Flotillin-1 antibody (F1180, 1:2000) was obtained from Sigma Aldrich. Mouse anti-Clathrin Heavy Chain antibody (610499, 1:1000) and mouse anti-Caveolin 1 antibody (610406, 1:1000) were purchased from BD Biosciences (Franklin Lakes, NJ, USA). Mouse anti-Alpha 1 Sodium Potassium ATPase antibody (ab7671, 1:2000) was purchased from Abcam (Waltham, MA, USA). Mouse Tetra His antibody (34670, 1:2000) was bought from QIAGEN (Germantown, MD, USA). The mouse anti-SLC22A1 antibody was obtained from Novus Biologicals (Littleton, CO, USA).

### 4.2. Cell Culture 

HEK Flp-In^TM^-293 (HEK293; Thermo Fisher Scientific, Waltham, MA, USA) cells that stably express His-tagged NTCP or untagged OCT1 were used and grown at 37 °C and 5% CO_2_. Cells were cultured in Dulbecco’s Modified Eagle’s Medium (DMEM) from American Type Culture Collection (ATCC, Manassas, VA, USA: 30-2002) supplemented with 10% fetal bovine serum (Hyclone, Logan, UT, USA), 100 U/mL penicillin, 100 μg/mL streptomycin, and 500 μg/mL hygromycin (Thermo Fisher Scientific). 

### 4.3. Transporter Uptake Assays 

NTCP-mediated uptake of [^3^H]-taurocholic acid and OCT1-mediated uptake of [^3^H]-MPP^+^ were measured using a previously established and published procedure [37,38]. HEK293 uptake buffer was used for all uptake solutions. When sodium-free conditions were required, 142 mM sodium chloride was replaced with choline chloride. The total concentration of taurocholic acid varied for the different experiments and can be found in the figure legends. Briefly, cells were washed with warm uptake buffer and then incubated with uptake solution containing either radiolabeled taurocholate or MPP^+^ at 37 °C for the selected amount of time. Uptake was halted by removing the radioactive uptake solution and immediately washing the cells with ice-cold uptake buffer. The cells were then lysed using 1% TX-100 in PBS. Radioactivity was determined using a beta counter, and uptake was normalized for total protein measured with the bicinchoninic acid (BCA) assay (Thermo Fisher Scientific). 

### 4.4. Cholesterol Addition, Depletion, and Quantification

HEK293 cells stably expressing NTCP or OCT1 were pretreated with Opti-MEM (control) or increasing concentrations of either cholesterol or MCD in Opti-MEM for 30 min. After treatment, radiolabeled taurocholate or MPP^+^ uptake was measured as described above. Free cholesterol was quantified using the Amplex^®^ Red Cholesterol Assay (Thermo Fisher Scientific), which produces a fluorescent signal corresponding to the amount of free cholesterol. In this assay, cholesterol is oxidized by cholesterol oxidase to H_2_O_2,_ which is then detected using Amplex^®^ Red reagent. If total cholesterol needs to be determined, esterified cholesterol can be converted to free cholesterol with cholesterol esterase.

### 4.5. Surface Biotinylation

HEK293 cells stably expressing EV, NTCP, or OCT1 were plated at 800,000 cells/well on 6-well poly-d-lysine coated plates. After two days in culture, cells were treated for 30 min with either Opti-MEM (control), cholesterol, or MCD. The medium was changed, and the cells were incubated for 15 min on ice. The assay then proceeded as previously described in [38].

### 4.6. Transporter Kinetics

Initial linear rates were determined for untreated and treated stably expressing HEK293 cells at low and high concentrations of taurocholate (0.1 and 200 μM) for NTCP and MPP^+^ (0.1 and 300 μM) for OCT1. Cells were plated on a 24-well poly-d-lysine coated plate at either 300,000 or 200,000 cells/well for NTCP and at either 200,000 or 100,000 cells/well for EV and OCT1 to perform an uptake after 24 or 48 h in culture. The control kinetics were measured after removing DMEM. Uptakes were performed using uptake buffer containing sodium chloride or choline chloride and measured at increasing concentrations of taurocholate or MPP^+^ at 15 s. The uptake was corrected for total protein and surface expression. Results were analyzed in GraphPad Prism 9 (GraphPad Software Inc., San Diego, CA, USA) (Michaelis–Menten kinetics).

### 4.7. Isolation of Lipid Rafts

A 15 cm plate of HEK293 cells expressing EV, NTCP, or OCT1 was washed with PBS and treated with Opti-MEM (control), cholesterol (0.8 mM for NTCP and 0.4 mM for OCT1), or MCD (5 mM for NTCP and 20 mM for OCT1) for 30 min. The cells were washed with PBS, and 1 mL of PBS was added to allow the cells to be scraped into an Eppendorf tube. The cells were centrifuged at 900 g_av_ for 5 min at 4 °C and then resuspended in a homogenization solution (1 mM NaCl, 250 μL of 5 mM Tris-HCl pH 7.43 in 50 mL ddH_2_O) containing protease inhibitors (Roche Diagnostics) before being homogenized in a glass Teflon homogenizer (20 strokes, handheld drill). The homogenate was centrifuged at 900 g_av_ for 10 min at 4 °C, after which the supernatant was collected and centrifuged at 10,000 g_av_ for 20 min at 4 °C. The final pellet was resuspended in 50 μL of homogenization solution, and 450 μL of 3% TX-100 in TNE (50 mM Tris Base, 150 mM NaCl, 2 mM Na_2_EDTA, pH 7.4) was added, and the samples were vortexed and incubated on ice for 30 min. The samples were then homogenized with a 25 G needle 20 times before mixing 1:1 with 80% sucrose in TNE (*w*/*v*). The 40% sucrose homogenate solution was then overlaid with a discontinuous 5–35% sucrose gradient (*w*/*v*, 4 mL of 35%, and 5 mL of 5% Sucrose) and centrifuged for 18 h at 197,000 g_av_. Samples were then collected in 1 mL aliquots using a Gilson Miniplus 2 peristaltic pump at 500.

### 4.8. Western Blotting

The surface biotinylation and sucrose gradient samples were heated for 10 min at 50 °C before separation using SDS-PAGE. Proteins were transferred to nitrocellulose membranes, and blots were blocked in 5% milk in TBS-T (TBS and 0.1% Tween 20) or 3% BSA in TBS-T when probing for OCT1. Blots were probed with primary antibodies against positive (caveolin and flotillin) and negative (clathrin and Na^+^/K^+^ ATPase) lipid raft markers in addition to NTCP and OCT1 overnight at 4 °C in the respective blocking solution. Blots were washed three times with TBS-T before incubation with HRP-conjugated secondary antibodies in 2.5% milk in TBS-T for 1 h at room temperature and washed four times before adding a chemiluminescent substrate. Signals were captured using the LI-COR Odyssey Fc (LI-COR, Lincoln, NE, USA) and quantified using Image Studio Lite Quantification Software.

### 4.9. Statistical Analysis

Data were analyzed for significant differences between groups using one-way ANOVA followed by “Dunnett’s multiple comparisons tests” in GraphPad Prism 9 (GraphPad Software Inc., San Diego, CA, USA). Significance was determined by a *p*-value less than 0.05.

## Figures and Tables

**Figure 1 ijms-23-08457-f001:**
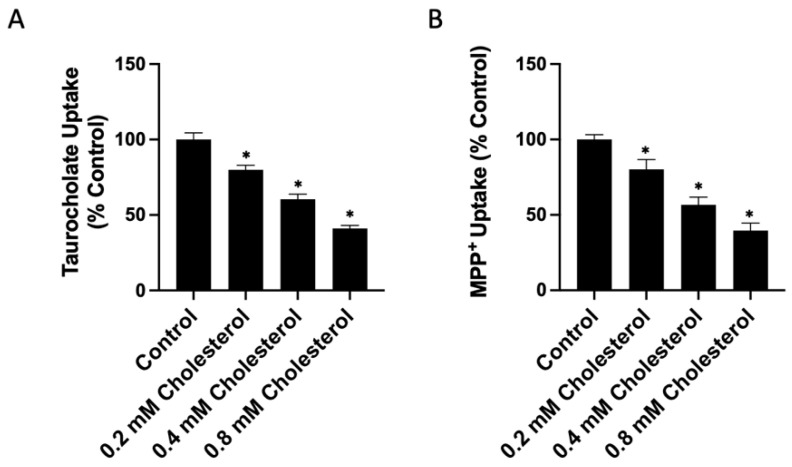
Transporter-mediated substrate uptake as a function of increased free cholesterol content. Uptake of (**A**) (100 µM) [^3^H]-taurocholate for NTCP- and (**B**) (16.7 nM) [^3^H]-MPP^+^ for OCT1-expressing HEK293 cells was measured for 30 s at 37 °C, 30 min after treatment with increasing cholesterol–MCD (cholesterol) concentrations. Net uptake was determined by subtracting the uptake measured in a sodium-free buffer from the uptake measured in a sodium-containing buffer for NTCP, or by subtracting uptake from empty-vector-transfected (EV) cells from OCT1-expressing cells for OCT1. Results were calculated as percent of control and reported as the mean ± SD of three independent experiments; asterisks indicate a *p* < 0.05.

**Figure 2 ijms-23-08457-f002:**
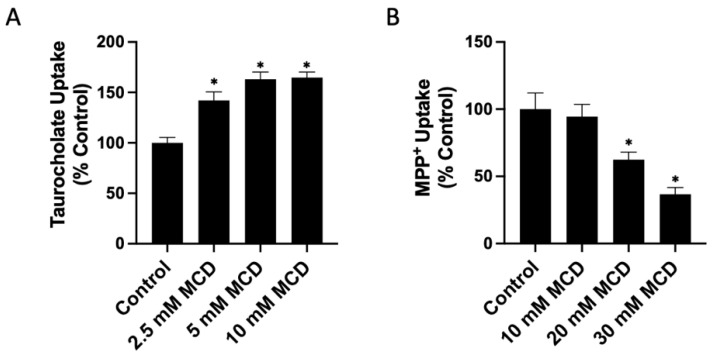
Transporter-mediated substrate uptake as a function of decreased free cholesterol content. Uptake of (**A**) (100 µM) [^3^H]-taurocholate for NTCP- and (**B**) (16.7 nM) [^3^H]-MPP^+^ for OCT1-expressing HEK293 cells was measured for 30 s at 37 °C, 30 min after treatment with increasing methyl-β-cyclodextrin (MCD) concentrations. Net uptake was determined as in Figure 1, and results were calculated as percent of control and reported as the mean ± SD of three independent experiments; asterisks indicate a *p* < 0.05.

**Figure 3 ijms-23-08457-f003:**
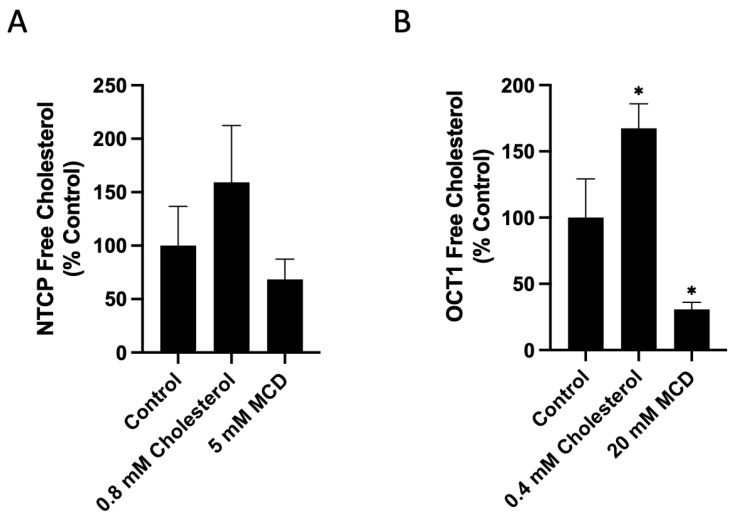
Free cholesterol content in transporter-expressing HEK293 cells. (**A**) NTCP- and (**B**) OCT1-expressing HEK293 cells were treated for 30 min with either Opti-MEM (control) or the indicated concentrations of cholesterol or MCD. The graphs represent the mean ± SD of 3 experiments for NTCP- and 4 experiments for OCT1-expressing cells; an asterisk indicates a *p* < 0.05 compared to the control group.

**Figure 4 ijms-23-08457-f004:**
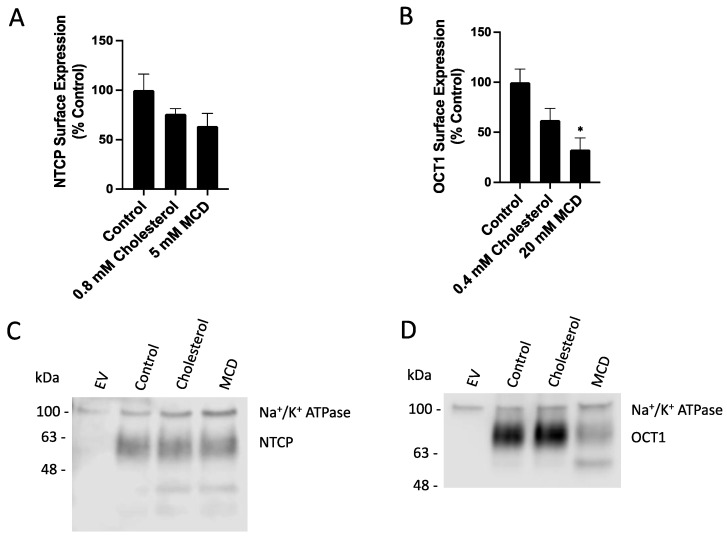
Surface expression of transporters after treatment with cholesterol or MCD. Quantification of plasma membrane expression for HEK293 expressing NTCP (**A**) or OCT1 (**B**). Cells were pretreated for 30 min with either Opti-MEM (control), cholesterol, or MCD. Representative western blots for NTCP (**C**) and OCT1 (**D**) surface expression are shown. Results were calculated as percent of control and reported as mean ± SEM of three independent experiments; an asterisk indicates a *p* < 0.05 compared to the control group.

**Figure 5 ijms-23-08457-f005:**
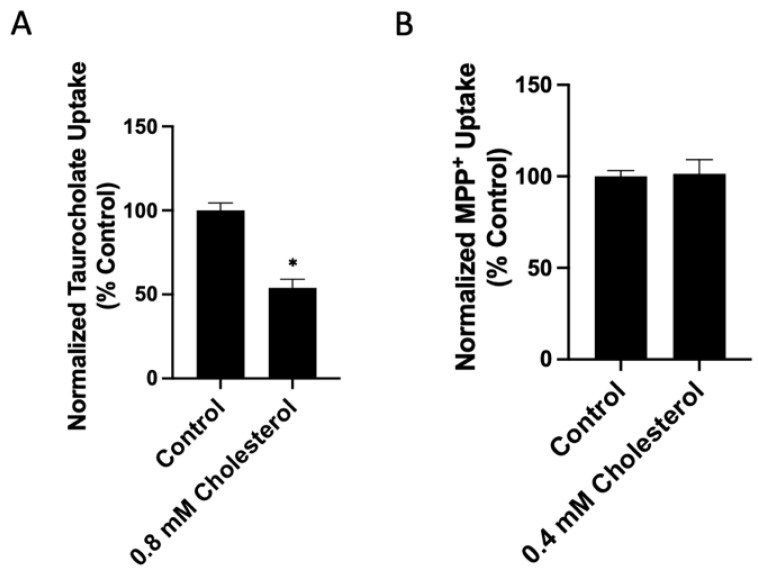
Normalized NTCP- and OCT1-mediated uptake after treatment with cholesterol. Uptake of (**A**) 100 µM [^3^H]-taurocholate by NTCP- and (**B**) 16.7 nM [^3^H]-MPP^+^ by OCT1-expressing HEK293 cells were corrected for surface expression. Net transporter-mediated uptake was calculated, and results are presented as percent of control; means ± SD of *n* = 3 in (**A**) or *n* = 4 in (**B**) are shown; asterisks indicate a *p* < 0.05.

**Figure 6 ijms-23-08457-f006:**
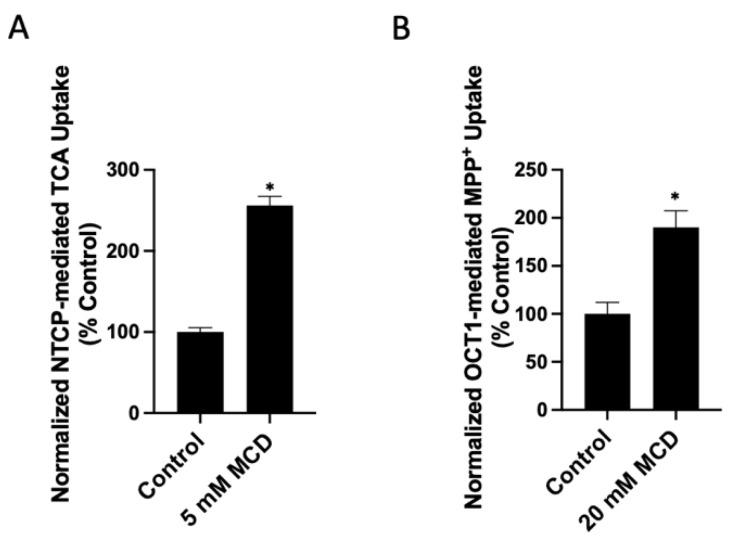
Normalized NTCP- and OCT1-mediated uptake after treatment with MCD. Uptake of (**A**) 100 µM [^3^H]-taurocholate by NTCP- and (**B**) 16.7 nM [^3^H]-MPP^+^ by OCT1-expressing HEK293 cells were corrected for surface expression. The net transporter-mediated uptake was calculated, and results are presented as percent of control; means ± SD of *n* = 3 in (**A**) or *n* = 4 in (**B**) are shown; asterisks indicate a *p* < 0.05.

**Figure 7 ijms-23-08457-f007:**
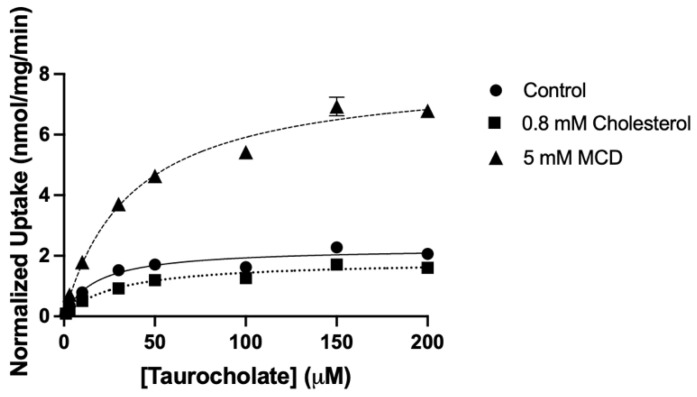
Normalized kinetics of NTCP-mediated taurocholate uptake after treatment with cholesterol or MCD. Uptake of increasing concentrations of taurocholate was measured at 37 °C for 15 s in NTCP-expressing HEK293 cells in a sodium-containing or sodium-free buffer, and corrected for protein. Net uptake was calculated by subtracting the uptake measured in choline chloride buffer from the uptake measured in a sodium chloride buffer and then correcting for NTCP surface expression. Values are the means ± SEM of at least three independent experiments performed in triplicates. Curves were fit with the Michaelis–Menten equation in GraphPad Prism.

**Figure 8 ijms-23-08457-f008:**
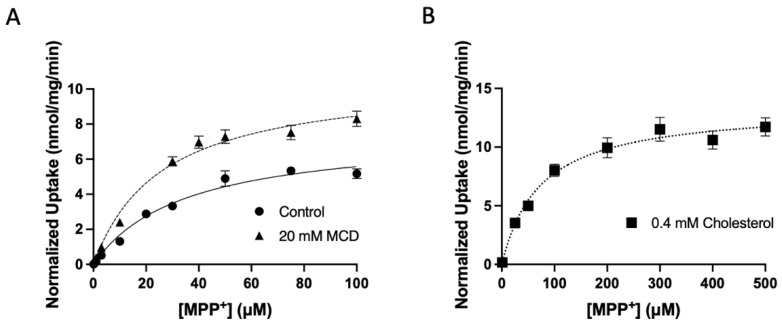
Normalized kinetics of OCT1-mediated MPP^+^ uptake after treatment with (**A**) MCD or (**B**) cholesterol. Uptake of increasing concentrations of MPP^+^ was measured at 37 °C for 15 s in EV and OCT1-expressing HEK293 cells in sodium-containing buffer and corrected for protein. Net uptake was calculated by subtracting the uptake measured in EV cells from the uptake measured in OCT1-expressing cells and then correcting for OCT1 surface expression. Values are means ± SEM of at least three independent experiments performed in triplicates. Curves were fit with the Michaelis–Menten equation in GraphPad Prism.

**Figure 9 ijms-23-08457-f009:**
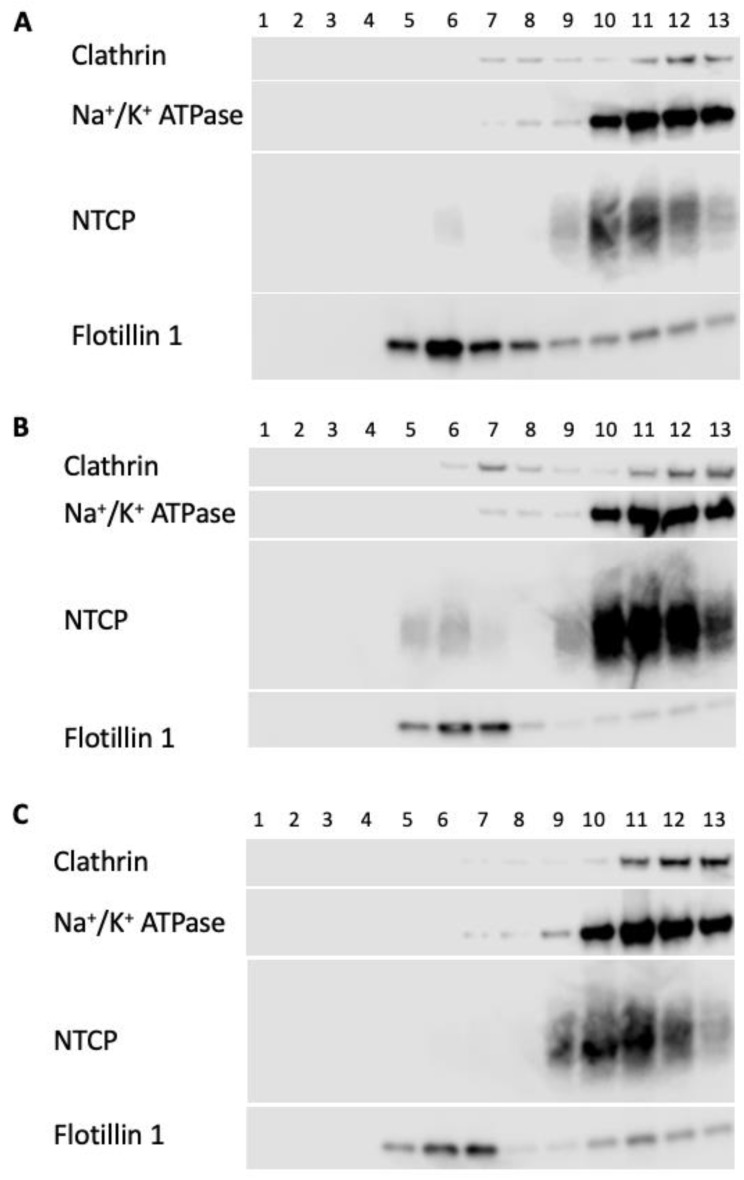
Distribution of marker proteins in lipid rafts isolated from NTCP-expressing HEK293 cells. NTCP-expressing HEK293 cells were treated with (**A**) Opti-MEM (control), (**B**) 0.8 mM cholesterol, or (**C**) 5 mM MCD, after which they were homogenized, centrifugated, and fractionated as outlined in the Materials and Methods section. Aliquots of the 1 mL gradient fractions were analyzed using western blotting. Positive lipid raft markers are caveolin and flotillin-1; negative markers are clathrin and Na^+^/K^+^ ATPase.

**Figure 10 ijms-23-08457-f010:**
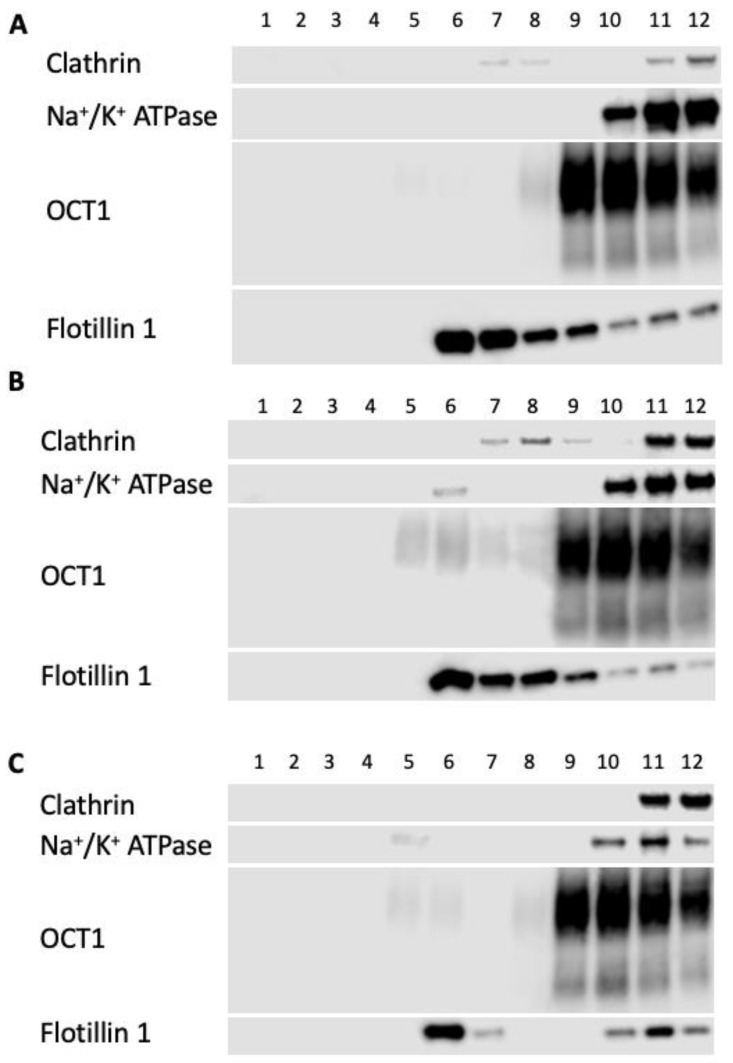
Distribution of marker proteins in lipid rafts isolated from OCT1-expressing HEK293 cells. HEK293 cells expressing OCT1 were incubated with (**A**) Opti-MEM (control), (**B**) 0.4 mM cholesterol, or (**C**) 20 mM MCD before being homogenized, centrifugated, and fractionated as outlined in the Materials and Methods section. Aliquots of the 1 mL gradient fractions were analyzed using western blotting. Positive lipid raft markers are caveolin and flotillin-1; negative markers are clathrin and Na^+^/K^+^ ATPase.

**Table 1 ijms-23-08457-t001:** Kinetic parameters after normalization for surface expression. Kinetic parameters, K_m_ and V_max_, were calculated using the Michaelis–Menten equation in GraphPad Prism and are reported as the average ± SEM of at least three independent experiments performed in triplicates; asterisks indicate a *p* < 0.05.

Transporter	Parameters	Control	Cholesterol	MCD
	K_m_ (µM)	18 ± 2.8	38 ± 28	39 ± 13
NTCP	V_max_ (nmol/mg/min)	2.3 ± 0.04	1.9 ± 0.1	8.3 ± 1.0
	V_max_/K_m_ (µL/mg/min)	132 ± 20	69 ± 37 *	224 ± 65 *
	K_m_ (µM)	39 ± 11	72 ± 17	26 ± 5
OCT1	V_max_ (nmol/mg/min)	7.6 ± 1.7	9.0 ± 2.3	7.2 ± 1.2
	V_max_/K_m_ (µL/mg/min)	209 ± 66	125 ± 26	279 ± 20 *

## Data Availability

All data are contained within the article.

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
