# Peer review of "Free Cholesterol Affects the Function and Localization of Human Na+/Taurocholate Cotransporting Polypeptide (NTCP) and Organic Cation Transporter 1 (OCT1)"

_ijms, 2022, doi:10.3390/ijms23158457_

Round 1
Reviewer 1 Report
Idowu et al present a biophysical characterization NTCP and OCT activation by cholesterol dependent trafficking of the proteins in and out detergent resistant membrane from HEK293T cells. The authors use a substrate transport assay to show activation of the transporters when cholesterol is removed and inhibition when cholesterol is increased. The authors confirm a change in membrane cholesterol using a fluorescent cholesterol assay and they measure the amount of transporter on the surface and adjust kinetic assay based on expression levels. Lastly they confirm movement of the transporters in and out of detergent resistant membranes (DRM) using western blots.
The experiments are well done and the conclusion are for the most part supported by their data. Cholesterol signaling and nanoscopic movement of transporters is an important emerging theme. There are a few weaknesses in the approach, they use some techniques that are crude and dated. There is one major concern that needs to be addressed prior to publication.
Major Point.
Point #1, Page 2 line 59 the authors state “The goal of the present study was to investigate whether increasing or decreasing membrane cholesterol levels would affect the function of and expression of liver uptake transporters NTCP and OCT1”. But cholesterol was already shown to affect the function of NTCP by Molina et al in 2008 and published in Biochim Biophys Acta[1]. Molina et al, used the exact same techniques as the current authors, HEK293 cells, DRMs, and MBCD depletion to show cholesterol regulates the function of NTCP. The authors need to lay out in the introduction all the data they are repeating on NTCP from previous publication. The introduction needs to explicitly say something like the follow: “NTCP was previously shown to increase activity when cholesterol was depleted”. The authors then need to put their results in context with the previous study. For example, “We saw a 2 fold increase in NTCP activity after cholesterol depletion. This was similar to the 2.5 fold increase previously seen” and then reference the study.
This type of acknowledgment needs to be applied regarding NTCP throughout the paper. Also please confirm that OCT1 has not been studied for its activity in response to cholesterol export and uptake. Provide at least one sentence towards the end of the introduction that helps the reader understand what studies the current authors are presenting that are new about NTCP. For example: “Here we account for surface expression of NTCP regulation by cholesterol and compare NTCP cholesterol regulation to that of OCT1”.
Minor point
Point #2. The amount of protein localizing with flotillin in high and low cholesterol is very modest (Figure 10B vs C). It is this reviewer’s opinion that the authors have interpreted the data correctly, despite the modest change. However, the use of detergent resistant membranes and methylbetacyclodextran (MBCD) are crude methods developed 20 years ago. It has been known for a long time that DRM’s are not the same as lipid rafts. [2] A major problem with DRMs is that proteins cholesterol regulates are out of lipid rafts due to the destruction of the cellular membrane in order to generate DRMs. Even small amounts of mechanical force on the plasma membrane can cause a protein to leave a lipid raft[3] and this has caused underestimation of transient proteins in the lipid raft. For example, the majority of amyloid precursor protein was found outside of lipid rafts by DRMs[4], but super res imaging showed it is mostly in lipid rafts with high cholesterol and moves out with low cholesterol[5].
Using super resolution imaging to look at intact cell membranes would be much more physiologically relevant and improve on the 2008 study. Nonetheless, if the authors lack access to this technology, they need to at least add a sentence to the discussion acknowledging that the DRMs may have under estimated the amount of transporter in lipid rafts due to the problems mentioned about.
Point #3 Pg1 line 11: Authors should use declarative statements where appropriate. For example, “We wanted to assess…” better reads “We assessed…”
Suggestions
Point #3. HEK293 cells have little relevance to hepatocytes. Why weren’t the DRM experiments done is a hepatic cell line with endogenously expressed transporter? I don’t think the liver cell will change the conclusions of the paper, so I don’t think this should be a point that has to be addressed with new experiments prior to publication. But if the primary author is still in the lab and can repeat a DRM experiment in hepatocytes, this would elevate the physiological relevance of the paper substantially.
Point #4 Cholesterol uptake and export with MBCD is crude and physiologically not very relevant. New techniques have evolved to export and uptake cholesterol using lipoproteins with and without FBS (e.g., apoE)[5]. ApoE is receptor mediated and the uptake and removal is physiological. Using a lipoprotein vs MBCD is not likely to change the conclusion of the paper, so unnecessary prior to publication, but if an experiment can easily be done, it would elevate the physiological relevance of the paper substantially.
References cited
1 Molina, H. et al. (2008) Localization of the Sodium-Taurocholate cotransporting polypeptide in membrane rafts and modulation of its activity by cholesterol in vitro. Biochim. Biophys. Acta - Biomembr. 1778, 1283–1291
2 Lichtenberg, D. et al. (2005) Detergent-resistant membranes should not be identified with membrane rafts. Trends Biochem. Sci. 30, 430–436
3 Petersen, E.N. et al. (2016) Kinetic disruption of lipid rafts is a mechanosensor for phospholipase D. Nat. Commun. 7, 13873
4 Ehehalt, R. et al. (2003) Amyloidogenic processing of the Alzheimer β-amyloid precursor protein depends on lipid rafts. J. Cell Biol. 160, 113–123
5 Wang, H. et al. (2021) Regulation of beta-amyloid production in neurons by astrocyte-derived cholesterol. Proc. Natl. Acad. Sci. 118, e2102191118
Author Response
Response to reviewer 1:
We thank the reviewer for the thoughtful comments and suggestions. We incorporated the suggestions as far as we could and believe that the quality of the manuscript has increased. We are not ready yet to perform the suggested experiments to be included in this manuscript but are working on cell lines that are closer to human hepatocytes and express the respective transporters. Our answers to the specific points the reviewer raised are given below in italics.
Major Point.
Point #1, Page 2 line 59 the authors state “The goal of the present study was to investigate whether increasing or decreasing membrane cholesterol levels would affect the function of and expression of liver uptake transporters NTCP and OCT1”. But cholesterol was already shown to affect the function of NTCP by Molina et al in 2008 and published in Biochim Biophys Acta[1]. Molina et al, used the exact same techniques as the current authors, HEK293 cells, DRMs, and MBCD depletion to show cholesterol regulates the function of NTCP. The authors need to lay out in the introduction all the data they are repeating on NTCP from previous publication. The introduction needs to explicitly say something like the follow: “NTCP was previously shown to increase activity when cholesterol was depleted”. The authors then need to put their results in context with the previous study. For example, “We saw a 2 fold increase in NTCP activity after cholesterol depletion. This was similar to the 2.5 fold increase previously seen” and then reference the study.
This type of acknowledgment needs to be applied regarding NTCP throughout the paper. Also please confirm that OCT1 has not been studied for its activity in response to cholesterol export and uptake. Provide at least one sentence towards the end of the introduction that helps the reader understand what studies the current authors are presenting that are new about NTCP. For example: “Here we account for surface expression of NTCP regulation by cholesterol and compare NTCP cholesterol regulation to that of OCT1”.
We have included a more direct comparison to the Molina et al. 2008 study that was published using rodent and not human NTCP throughout the manuscript and tried to clarify what was similar and what we did in addition to the Molina et al. 2008 paper.
Minor point
Point #2. The amount of protein localizing with flotillin in high and low cholesterol is very modest (Figure 10B vs C). It is this reviewer’s opinion that the authors have interpreted the data correctly, despite the modest change. However, the use of detergent resistant membranes and methylbetacyclodextran (MBCD) are crude methods developed 20 years ago. It has been known for a long time that DRM’s are not the same as lipid rafts. [2] A major problem with DRMs is that proteins cholesterol regulates are out of lipid rafts due to the destruction of the cellular membrane in order to generate DRMs. Even small amounts of mechanical force on the plasma membrane can cause a protein to leave a lipid raft[3] and this has caused underestimation of transient proteins in the lipid raft. For example, the majority of amyloid precursor protein was found outside of lipid rafts by DRMs[4], but super res imaging showed it is mostly in lipid rafts with high cholesterol and moves out with low cholesterol[5].
Using super resolution imaging to look at intact cell membranes would be much more physiologically relevant and improve on the 2008 study. Nonetheless, if the authors lack access to this technology, they need to at least add a sentence to the discussion acknowledging that the DRMs may have under estimated the amount of transporter in lipid rafts due to the problems mentioned about.
We agree with the reviewer that our techniques are rather outdated but we currently do not have access to a confocal microscope with a resolution that would allow us to perform the super-resolution imaging.
Point #3 Pg1 line 11: Authors should use declarative statements where appropriate. For example, “We wanted to assess…” better reads “We assessed…”
We thank the reviewer for this suggestion. We have used Grammarly to improve English in this manuscript and hope that the text now reads better.
Suggestions
Point #3. HEK293 cells have little relevance to hepatocytes. Why weren’t the DRM experiments done is a hepatic cell line with endogenously expressed transporter? I don’t think the liver cell will change the conclusions of the paper, so I don’t think this should be a point that has to be addressed with new experiments prior to publication. But if the primary author is still in the lab and can repeat a DRM experiment in hepatocytes, this would elevate the physiological relevance of the paper substantially.
We agree with the reviewer that HEK293 cells are not liver cells or comparable to human hepatocytes. We are currently establishing HepG2 cell lines that express NTCP and OCT1 and plan to use these cell lines in the future to expand the current findings and hopefully even get access to the confocal microscope to perform the super-resolution imaging with the HepG2 cells. In our opinion, HepG2 cells are closer to human hepatocytes and once we establish the expression of the respective transporters should allow getting more relevant results than just from HEK293 cells.
Point #4 Cholesterol uptake and export with MBCD is crude and physiologically not very relevant. New techniques have evolved to export and uptake cholesterol using lipoproteins with and without FBS (e.g., apoE)[5]. ApoE is receptor mediated and the uptake and removal is physiological. Using a lipoprotein vs MBCD is not likely to change the conclusion of the paper, so unnecessary prior to publication, but if an experiment can easily be done, it would elevate the physiological relevance of the paper substantially.
We agree with the reviewer that changes in free cholesterol as we induced them with MBCD is a rather harsh treatment. We have planned to extend our findings, as soon as the above-mentioned HepG2 cell lines are established, and use the methods suggested by the reviewer to study chronic rather than acute changes in free cholesterol using more physiological methods to change free cholesterol levels.
Reviewer 2 Report
I have read the paper by Idowu and Hagenbuch, which assesses that the modulation of plasma cholesterol content is able to affect OCP1 and NCPT functions.
The manuscript could be interesting, but this reviewer has several concerns:
1) Authors should state in the paper the reason why they used HEK cells as experimental model, They start the paper describing hepatic pathologies and transporters, while they use kidney derived cells. This is not clear and confunds the reader.
2) Authors always speak about MEMBRANE cholesterol content. How did they measure it? The Amplex red cholesterol kit, as far as I know, is used for serum/plasma cholesterol content evaluation and, in any case, it could evaluate the total free cholesterol content extracted by the cells. This point must be clarified and the statements MEMBRANE CHOLESTEROL corrected. Moreover, the method has to be described in M&M section. I did not find it.
3) Figure legends should be more precise, moreover statistical analyses as well as the N of the experiments should be always written in them.
4) A paragraph describing statistical analysis is completely missing in M&M section. Which kind of statistical analysis has been performed? This is a very important concern!
Author Response
Response to reviewer 2:
We thank the reviewer for the thoughtful comments and suggestions. We hope that we incorporated them appropriately and believe that the quality of the manuscript has increased and methods are described as complete as possible. Our answers to the specific points the reviewer raised are given below in italics.
1) Authors should state in the paper the reason why they used HEK cells as experimental model, They start the paper describing hepatic pathologies and transporters, while they use kidney derived cells. This is not clear and confunds the reader.
We agree with the reviewer that in an ideal case all the experiments would have been performed with human hepatocytes. However, human hepatocytes are difficult to get in good quality and quantity and HEK293 cells have been extensively used to characterize transporters. We plan to establish HepG2 cells that express NTCP and OCT1 and then use those cell lines to extend our studies to investigate chronic instead of acute effects of cholesterol.
2) Authors always speak about MEMBRANE cholesterol content. How did they measure it? The Amplex red cholesterol kit, as far as I know, is used for serum/plasma cholesterol content evaluation and, in any case, it could evaluate the total free cholesterol content extracted by the cells. This point must be clarified and the statements MEMBRANE CHOLESTEROL corrected. Moreover, the method has to be described in M&M section. I did not find it.
We apologize that we forgot to include the method of the cholesterol assay in the Materials and Methods section, and we added the method. The assay only determines free and not total cholesterol. If total cholesterol needs to the measured, all the esterified cholesterol needs to be converted to free cholesterol with the enzyme cholesterol esterase that is included in the kit. However, we did measure cholesterol in the absence of the esterase and thus only measured free cholesterol. Furthermore, the majority of free cholesterol is found in the cell membrane and therefore we keep saying “membrane free cholesterol”. We have added a sentence in the manuscript clarifying this.
3) Figure legends should be more precise, moreover statistical analyses as well as the N of the experiments should be always written in them.
We are sorry that the reviewer did not see that the N as well as the p-values were included in the figure legends where statistics were performed.
4) A paragraph describing statistical analysis is completely missing in M&M section. Which kind of statistical analysis has been performed? This is a very important concern!
We apologize for omitting the description of the statistics. We have added the following description to the Materials and Methods section.
4.9. Statistical Analysis
Data were analyzed for significant differences between groups using one-way ANOVA followed by “Dunnett’s multiple comparisons tests” in GraphPad Prism 9 (GraphPad Software Inc., San Diego, CA). Significance was determined by a p-value less than 0.05.
Round 2
Reviewer 2 Report
In my opinion the paper has been improved and can be published.